# The Impact of In-Event Health Services at Europe’s Largest Electronic Dance Music Festival on Ems and Ed in the Host Community

**DOI:** 10.3390/ijerph20043207

**Published:** 2023-02-11

**Authors:** Kris Spaepen, Robby Cardinas, Winne A. P. Haenen, Leonard Kaufman, Ives Hubloue

**Affiliations:** 1Research Group on Emergency and Disaster Medicine, Vrije Universiteit Brussel, 1050 Brussels, Belgium; 2Het Vlaamse Kruis, 2630 Antwerp, Belgium; 3Crisis Management at Federal Public Health Service, 2000 Antwerp, Belgium

**Keywords:** emergency medical services, emergency department, mass gatherings, mass gathering medicine, mitigation, electronic dance music festival, event medicine

## Abstract

Background: Electronic dance music festivals (EDMF) can cause a significant disruption in the standard operational capacity of emergency medical services (EMS) and hospitals. We determined whether or not the presence of in-event health services (IEHS) can reduce the impact of Europe’s largest EDMF on the host community EMS and local emergency departments (EDs). Methods: We conducted a pre-post analysis of the impact of Europe’s largest EDMF in July 2019, in Boom, Belgium, on the host community EMS and local EDs. Statistical analysis included descriptive statistics, independent *t*-tests, and χ^2^ analysis. Results: Of 400,000 attendees, 12,451 presented to IEHS. Most patients only required in-event first aid, but 120 patients had a potentially life-threatening condition. One hundred fifty-two patients needed to be transported by IEHS to nearby hospitals, resulting in a transport-to-hospital rate of 0.38/1000 attendees. Eighteen patients remained admitted to the hospital for >24 h; one died after arrival in the ED. IEHS limited the overall impact of the MGE on regular EMS and nearby hospitals. No predictive model proved optimal when proposing the optimal number and level of IEHS members. Conclusions: This study shows that IEHS at this event limited ambulance usage and mitigated the event’s impact on regular emergency medical and health services.

## 1. Introduction

For the definition of a mass gathering event (MGE), we turn to Arbon [1], who defines an MGE as an event “where there is the potential for a delayed response to [health] emergencies because of limited access to patients or other features of the environment and location”. The World Health Organization adds that an MGE “attracts sufficient numbers of people to strain the planning and response resources of the community, city, or nation hosting the event” [2]. Electronic dance music festivals (EDMF) form a specific subset of MGE. They are described by Fitzgibbon [3] as “live music events that feature multiple performers and are often longer than conventional concerts, in some cases days, in duration”.

Providing health care at MGEs comes with several challenges, mainly related to patients’ exit and the environment of the MGEs [4]. Electronic dance music festivals, however, pose even more challenges, with EDMF primarily being organized during hot summer months, attracting an active, young, and mobile crowd who engage more in alcohol and illicit drug use (e.g., 3,4-methylenedioxymethamphetamine (MDMA), cocaine, gamma-hydroxybutyric acid (GHB), and ketamine) [3,5,6,7,8,9]. The mass gathering literature remains superficial on studies of MGE’s impact on the standard operational capacity of emergency medical services (EMS) and emergency departments (EDs) in the MGE host community. This is worrying because Heiby [10] showed that an MGE causes a significant increase in the utilization of EMS, despite it being known that in-event health services (IEHS) are an effective means of preventing an influx of patients in EDs [11,12,13]. With sufficient onsite clinical resources and the skill set of in-event health professionals, they could either treat the patients onsite and either release them to the event or hold them at the medical post for a short observation period (for a maximum of two hours). This “hold and treat model” adopted by IEHS could minimize the disruption of the standard functional capacity of EMS and EDs in the host community.

However, it is essential to have a valid predictive model for patient presentations to predict the medical workload and the corresponding number and level of IEHS.

As a recent study showed, the Belgian Plan Risk Manifestation (PRIMA) model is a valid prediction tool for predicting patient presentations at music MGEs in Belgium [9]. The PRIMA model also generates advice and recommendations for the local government on the amount and level of IEHS at the MGE. The use of this model provides MGE planners and governments with invaluable information on the levels of IEHS needed at the MGE to ensure good health outcomes for attendees. Literature on whether such a predictive model mitigates the impact of the MGE on the standard operational capacity of EMS and hospitals in the host community is scarce. The focus of this study was not on evaluating the performance of the PRIMA model but aimed to describe if the amount and level of onsite IEHS based on the PRIMA model mitigated the impact of Europe’s largest EDMF on local EMS and hospitals.

## 2. Materials and Methods

### 2.1. Study Design and Setting 

This pre-post study was conducted at one outdoor EDMF that took place spread over the two last weekends in July 2019. The festival took place in Boom, Belgium. Boom has approximately 18,000 inhabitants. The festival is organized annually during the last two weekends of July. The festival commenced on Friday at noon and ended at 01:00 h on Saturday and Sunday. The final festival day was Sunday, when the music stopped at midnight on Monday. In the neighboring village of Rumst (15,000 inhabitants), near the festival site is a camping site where 35,000 visitors stayed. The camping site opened on Thursday at midday and closed on Monday at 16:00 h. For this study, we counted the days the campground was open. All 400,000 tickets were sold out within minutes for the two weekends.

The MG was bound and ticketed, and attendees could purchase alcohol on site. Potable water was freely available at water dispensers around the festival and camping site. The event site was a recreation area of 0.75 square kilometers (including the campground for 35,000 visitors). The festival site contained various stages—some even on pontoons on clay pits—with the main stage holding up to 25,000 people. The entrance and exit of 66,000 visitors isolated the event site during the event. Visitors entered or exited the festival site via two separate gates. One gate was for single-day tickets and was located on the south side of the festival site. The other gate was located on the north side of the festival site and was the entrance or exit point for the visitors who stayed at the camping site.

### 2.2. Plan Risk Manifestation (PRIMA) Model

The Plan Risk Manifestation (PRIMA) model is a Belgian risk analysis tool that contains a calculation model. The model is based on three variables. All numbers were calculated by studying the number of emergencies (life-threatening, non-life-threatening, and minor ailments and injuries) regulated and dispatched by Belgian emergency dispatch centers. First is the number of life-threatening emergencies. This variable was set to 0.019/10,000/5 h. The second variable is the number of non-life-threatening emergencies and was set at 0.19/10,000/5 h. The final variable, minor complaints and injuries, were set at 2/10,000/5 h.

Adding to these variables, the model has three medical risk axes (i.e., isolation risk, population risk, and risk at illness) [14]. The isolation risk considers the time EMS needs to reach the victim. Following Belgian legislation, a delayed response for EMS due to the MG justifies EMS or Basic Life Support (BLS) crews on site. If the threshold for the arrival of a mobile emergency group (consisting of at least one emergency nurse and one emergency physician) on site would exceed 15 min due to the MGE, the presence of at least one mobile emergency group at the event is advised. The population risk refers to the increased population during the MGE. In Belgium, regular EMS provides 1 ambulance per 21,000 inhabitants, so when the number of people attending an MGE exceeds 10,000, the PRIMA model advises an additional ambulance. The risk at illness takes the possible excess of illness at a specific type of MGE compared to the baseline Belgian population. When this risk is perceived as real, precautionary measures must be taken by IEHS.

### 2.3. Calculation Model of the PRIMA Model for the Number and Level of Staffing

When at least one ambulance is needed according to the population risk, the number of the population risk is given as advice. When the probability of ambulance transport is greater than one transport in 2 h, an ambulance must also be placed. When the likelihood of a patient needing transport is lower than one in every two hours, and no ambulance is required according to the population risk, an ambulance should still be placed when the distance to the nearest ambulance is more than 15 min.

When at least one mobile emergency group is needed according to the population risk, the number of the population risk is given as advice. When no mobile emergency group is required according to population risk, a mobile emergency group should still be placed when the risk of a life-threatening emergency is more significant than one in three hours (i.e., according to disease risk). When no mobile emergency group is required according to population risk and the risk of a life-threatening condition is less than one in three hours, a mobile emergency group should not be provided if the population is less than 10,000 visitors. When previous requirements are not met, a mobile emergency group must still be provided if the nearest mobile emergency group is more than a twenty-minute drive away. According to the Belgian Federal Public Health Service, based on previous, unpublished studies, a nurse is used in 80% of the patients who need to be transported to the hospital and 40% of the patients who need to be cared for at the medical posts. A physician is used in 80% of the patients that need to be transported to the hospital and in 20% that require first aid care.

Following Belgian guidelines on EMS response times and the above risk factors, the calculation model predicts the number of patient presentations. This prediction advises the level and number of onsite medical personnel. Preventing disruption of regular EMS and mitigating a potential surge on nearby ED’s is the rationale behind the advice produced by PRIMA. Extensive detail on the variables and parameters used to predict the number of patient presentations can be found in the specific publication [14]. The advice generated by PRIMA for this specific MGE can be found in Table 1.

### 2.4. Organization of In-Event Health Services

The Flemish Cross provided in-event health services, and IEHS was available to attendees in a staged approach (Figure 1). Within this staged approach, IEHS was freely available to all attendees. All medical posts (festival and camping site) were marked and recognizable by flags and signage. Upon arrival at the medical post, an emergency nurse performed a preliminary triage to assign patients to the appropriate treatment zone. The patients were then helped by either first-aid responders or health professional volunteers.

To mitigate the impact of the MGE on regular emergency departments in the region, IEHS maximized its efficiency by organizing six medical posts—three on the festival site during festival hours and three at the camping site, irrespective of festival hours. Onsite medical care was provided by first-aid responders, paramedics, emergency nurses, and emergency physicians.

Each medical post at the festival site had at least 38 first-aid responders, and paramedics, all trained at basic life support (BLS) level to provide in-event first-aid care. At least two emergency nurses and two emergency physicians per medical post provided in-event professional care. As in Belgium, most paramedics are trained at BLS level; IEHS provided two onsite, dedicated mobile emergency groups (consisting of an emergency nurse and an emergency physician) to help paramedics during transport of patients with altered or depressed levels of consciousness or a compromised airway.

All medical posts at the camping site had an equivalent number of first aid responders, paramedics, and logistic support to provide in-event first aid. All posts at the camping site had at least one emergency nurse and six emergency nurses in total during the nighttime (two per medical post), accompanied by three emergency physicians during the day and four during the night. At the festival site, IEHS provided one dedicated mobile emergency group to assist with transporting critical or unstable patients.

All medical posts (at the festival and camping site) had sufficient equipment to provide first aid care (BLS level) as well as a professional health care level (advanced life support) such as suturing, splinting, defibrillation, medications (the content of medicines is added as an additional file), rapid sequence intubation, or surgical airway, intravenous or intra-osseous line insertions. Imaging services (i.e., X-rays) were unavailable to the onsite medical teams.

Patients were transported to the hospital only when a physician deemed it necessary. Transports (accompanied or not by a medical team) were performed with standby ambulances provided by IEHS. Physicians did not use specific criteria or guidelines to decide whether or not a patient had to be transported to a nearby hospital. Receiving hospitals were pre-alerted by the referring physician on the patient’s clinical situation. The organization of the in-event health services was based on the advice of the Plan Risk Manifestations (PRIMA) model, and no resources were deployed from regular EMS to support IEHS.

### 2.5. Organization of Regular EMS and Emergency Departments in the Region of the MGE

Emergency medical services (out-of-hospital emergency services) start with an emergency dispatch center collecting a request for medical assistance by telephone, handling and organizing the response by dispatching the nearest most suitable ambulance available. The emergency rescue zone Rivierenland (i.e., firefighting stations of Boom and Willebroek) operate EMS ambulances in the region around the festival site.

The ambulance of Boom is operational between 08:00 h and 18:00 h. On average, the ambulance of Boom has a mean number of 2.39 EMS responses per 24 h. On weekend days (Friday, Saturday, and Sunday) during the summer months of 2019 (June, July, and August). The ambulance in the neighboring town of Willebroek is operational 24/7 (covering for Boom between 18:00 h and 08:00 h). On average, the ambulance in Willebroek has a mean number of 5.30 EMS responses on weekend days (Friday, Saturday, and Sunday) during the summer months of 2019 (June, July, and August). During the festival weekend, from Friday until Monday, the ambulance in Boom is active 24/7.

In Belgium, we find approximately 1.24 emergency departments per 100,000 residents, which is exceptionally high. According to the Belgian Health Care Knowledge Center (2016), there were an average of 290 emergency contacts per 100,000 residents in 2012, which is higher than in surrounding countries (e.g., Denmark, the Netherlands, France, and England) [15]. In Belgium, one can find two types of emergency departments. Depending on staffing requirements, infrastructure, and functional norms, a distinction is made between specialized and non-specialized emergency departments. Specialized emergency departments ought to be able to “*secure, stabilize and restore the vital functions*” and are “*responsible for the care of anyone who presents himself or is brought to the service with a health condition that can or may require immediate care*” [16]. Non-specialized emergency departments, on the other hand, only deal with first care and treatment of patients with an acute pathology.

There are five hospitals with a specialized ED within a fifteen-kilometer radius of the event [17]. The nearest hospital with a specialized ED is on the outskirts of the neighboring town of the festival and camping site-in Rumst—a mere three kilometers from the event site. It is a relatively small specialized emergency department in a general hospital with just over 200 beds, with a mean daily number of ED admissions of 52.35 patients per 24 h. Eleven kilometers from the event site are Antwerp’s University Hospital (573 beds), with a mean daily number of ED admissions of 96.21 patients per 24 h, and the general hospital of Mechelen (643 beds), with a mean daily number of ED admissions of 85.71 patients per 24 h. Fifteen kilometers from the event site are GZA Sint Augustinus hospital (568 beds), with a mean daily number of ED admissions of 78.90 per 24 h, and ZNA Middelheim hospital (705 beds), with a mean daily number of ED admissions of 120.38 patients per 24 h.

### 2.6. Population and Sample

Over the two weekends, approximately 400,000 people attended the EDMF. We included all patients who presented themselves for medical care at one of the onsite medical posts (both at the festival and camping site) in our sample. We defined patient presentations as all assessments by in-event healthcare personnel (from first aid-responders to complete patient assessments by emergency physicians).

### 2.7. Data Collection

In-event health services (The Flemish Cross) collected all data from the actual patient presentations. This IEHS organization uses its specific patient registration software, including patient demographics, triage code, administered treatment, and patient disposition. In-event health services handed anonymized data to the lead researcher in a Microsoft Excel^®^ (Microsoft Corporation, Richmond, VA, USA) spreadsheet.

The spreadsheet obtained from IEHS included the patient’s gender, the main reason for visiting IEHS, triage code given to patients by the triage nurse using the Simple Triage And Rapid Treatment disaster triage tool [18]. It also contained the physician’s diagnosis (when seen by an emergency physician), their ensuing treatment requirements, and if patients needed transport to local hospitals. We did not use a specific data dictionary (definitions) regarding the collected codes because of the lack of detailed information handed to the researchers by the IEHS.

Along with demographical data (except patients’ gender), no clinical data (e.g., diagnosis and treatment) were obtained because of privacy restrictions. We did not encounter missing data. We obtained meteorological data (temperature in °Celsius, humidity in %) through the Royal Meteorological Institute of Belgium. We gathered attendance estimates from the event managers (based on official ticket sales), and we obtained data on EMS ambulance interventions through emergency rescue zone Rivierenland. Data on the ED visits and hospital admissions were obtained through the head nurses or medical directors of the EDs of the respective hospitals.

### 2.8. Statistical Analysis

The statistical analysis includes descriptive statistics such as frequency distributions, means, and medians. As Lund [19] described, we divided the number of patients by the number of attendees at the event for the patient presentation rate (PPR) and divided the number of patients transported from the medical posts to the hospital’s ED divided by the number of attendees for the transport to hospital rate (TTHR). Turris et al. [20] described that the number of attendees was divided by the number of people in the host community to calculate the event-to-host population ratio (EHP). To assess whether the number and level of IEHS were optimal, we compared the actual number of in-even medical staff to predictions based on existing literature.

We collected the number of calls for the ambulances of Boom and Willebroek for June, July, and August, on Friday, Saturday, and Sunday in 2017, 2018, and 2019. We ran independent *t*-tests to compare the means of EMS interventions of the ambulance of Boom and the ambulance of Willebroek when there is no MGE compared to when there is a MGE. We calculated the odds ratios to determine whether the need for transport to hospital was bigger for those who were treated by first aid personnel compared to patients who were treated by health care professionals. To assess the contribution of each of the reasons for needing transport after consulting IEHS to the impact of the MGE on in-event and emergency department services, we ran a χ^2^ analysis. We used GraphPad Prism software version 9.3 (GraphPad Software, La Jolla, CA, USA, www.graphpad.com) (accessed on 3 November 2021) for the statistical tests. We set confidence interval levels at 95% and deemed results statistically significant if *p* < 0.05.

## 3. Results

Over the festival period (ten days), 12,451 (3.11% [95% CI 3.06 to 3.16]; 31.13/1000) visitors of the festival attended IEHS for medical help. With 400,000 attendees in a host community of 33,000, the EHP is 12.12. Coinciding with the daily peak temperatures (Table 2), the majority of patients presented themselves during the two-hour period of 16.00 h–18.00 h (*n* = 1813; 14.6% [95% CI 13.97 to 15.23]) (Figure 2).

Males formed the majority of patients (*n* = 6604; 53% [95% CI 52.11 to 53.89]). In-event health services were mostly consulted for trauma-related injuries (*n* = 6795; 54.06% [95% CI 53.71 to 55.49]). The most common reasons for seeking medical care were wound care because of blisters (*n* = 1764; 14.2% [95% CI 13.57 to 14.83]), wound care because of lacerations or abrasions (*n* = 1536; 12.3% [95% CI 11.71 to 12.89]), and headache (*n* = 925; 7.4% [95% CI 6.93 to 7.87]). More data on gender, initial triage code, level of care, and discharge disposition over the festival period are displayed in Table 3.

Life-threatening emergencies were rare. Only 120 patients (1.0% [95% CI 0.82 to 1.18]) were thought to need advanced life support measures. One patient was resuscitated on the scene and later died in hospital. Over the MGE period, only 152 patients (1.22% [95% CI 1.20 to 1.24]) needed to be transported by IEHS to nearby hospitals.

### 3.1. In-Event First Aid Care Only

Most patient presentations required in-event first aid only (*n* = 11,600; 93.17% [95% CI 92.72 to 93.62]). These were patients with triage code T3 (minor). The most common injury was blisters (*n* = 1760; 15.2% [95% CI 14.5 to 15.8]), and the most prevalent illness was headache (*n* = 900; 7.8% [95% CI 7.2 to 8.3]). Although most of these patients presented to IEHS between 16:00 h and 18:00 h (*n* = 1718; 14.81% [95% CI 14.15 to 15.47]), patients with minor acuity sought medical attention throughout the whole event (Figure 3).

Patients were predominantly male (*n* = 6095; 52.5% [95% CI 51.6 to 53.5]). Patients who had in-event first aid only needed transport to hospital by IEHS in very few cases (*n* = 46; 0.4% [95% CI 0.38 to 0.42]). The main reason for patients who were initially seen by in-event first aid only to be transported to hospital was to have X-rays for suspected fractures (*n* = 13; 28.3% [95% CI 28.17 to 28.43]). Eight patients (0.1% [95% CI 0.04 to 0.16]) who received in-event first aid only went to the hospital by their own means.

### 3.2. In-Event Health Professional Care

Of all patients seen or treated, 851 (6.8% [95% CI 6.35 to 7.25]) were referred to in-event health professional care by the triage nurse. These were patients with triage codes T1 (immediate) and T2 (urgent). The most common reason to be referred to in-event health professionals was substance and/or alcohol intoxication (*n* = 280; 32.9% [95% CI 29.6 to 36.0]) (Table 4). Patients who were seen or treated by in-event health professionals were predominantly male (*n* = 509; 59.8% [95% CI 56.7 to 63.1]) and presented themselves or were brought to in-event health professionals between 21:00 h and 22:00 h (*n* = 81; 9.5% [95% CI 7.5 to 11.4]) and after 00:00 h and 01:00 h (*n* = 73; 8.6% [95% CI 7.0 to 10.4]).

Of the 851 T1 or T2 patients, 745 (87.54% [95% CI 87.32 to 87.76]) returned to the festival after receiving care. However, IEHS transported 106 (12.46% [95% CI 12.10 to 12.82]) T1 or T2 patients to nearby hospitals, of which the majority (*n* = 19; 18.3% [95% CI 17.7 to 18.9]) needed transport because of substance and/or alcohol intoxication. Nine (1.1% [0.5 to 1.8]) other patients who received care from in-event health professionals went to the hospital by their own means. We found a significant association (χ^2^ (1) = 956.2, *p* < 0.0001) when looking at the group of patients who needed to be transported to hospital by IEHS (*n* = 152) and the level of IEHS (in-event first aid or in-event health professional care). Based on the odds ratio, the odds of needing transport for patients treated by health care professionals were 35.74 (95% CI 25.08 to 50.77) times higher than those treated by first aid personnel.

Of the 851 patients seen by in-event health professionals, a subgroup of 120 (14.1% [95% CI 14.04 to 14.14]) patients had a potentially life-threatening condition (e.g., unconscious, compromised airway, apnea, hyperthermia, etc.), with nearly half of these patients presenting with substance and/or alcohol intoxication (*n* = 54; 45% [95% CI 44.9 to 45.1]). Thirty-five patients (29.2% [95% CI 29.1 to 29.3]) with a potentially life-threatening condition received resuscitative care in the onsite medical posts (e.g., airway management with oropharyngeal airway, endotracheal intubation, advanced life support resuscitation for cardiac arrest). All 35 patients were transported by IEHS, accompanied by an emergency physician, an emergency nurse, and two paramedics during transport to the hospital. One patient later died in the hospital after being resuscitated onsite by IEHS.

Of these 35 patients who needed resuscitative measures, once again, the most prevalent health problem was substance and/or alcohol intoxication (*n* = 10; 28.6% [95% CI 28.4 to 28.8]). With ten intoxicated patients in life-threatening condition and need of transport by IEHS, there was a significant association between intoxicated patients and whether they would need transport, χ^2^ (1) = 5.39, *p* = 0.02. However, based on the odds ratio, the odds of being treated onsite by IEHS and preventing transport to the hospital while being substance and/or alcohol intoxicated were 2.68 times higher than being intoxicated and needing transportation to a hospital.

### 3.3. Transport to Hospital by IEHS

Of all patients seen or treated by IEHS, only 152 (1.22% [95% CI 1.20 to 1.24]) needed transport by IEHS to nearby hospitals, resulting in a TTHR of 0.38/1.000 attendees. The three main reasons for needing transport were, in descending order, suspected fractures or orthopedic trauma (*n* = 37; 24.3% [95% CI 24.16 to 24.44]) followed by substance and/or alcohol intoxication (*n* = 30; 19.7% [95% CI 19.55 to 19.85]) and non-specified traumatic injuries (*n* = 21; 13.8% [95% CI 13.66 to 13.96]). Most transports were performed between 23:00 h and 02:00 h (*n* = 29; 19.07% [95% CI 18.92 to 19.22]). Thirty-five patients (23.03%; 95% CI [22.89 to 23.17]) needed to be accompanied by an emergency physician and emergency nurse because of a potentially life-threatening condition (altered or depressed levels of consciousness or compromised airways). Seventeen patients who went to a hospital by their own means were not considered in the following tests.

### 3.4. In-Event Number and Level of Staffing

To assess whether the number and level of IEHS were optimal, we compared the actual number of in-event medical staff to predictions based on existing literature and the PRIMA model. When proposing the optimal number of physicians on site, the model by Krul [20,21] proposed 12.5% under the actual number of physicians on site. The PRIMA model proposed 12.5% more than the exact number deployed at the MGE. The model by Arbon [22] proposed the least physicians on site, with 87.5% fewer physicians than were on site.

When proposing the optimal number of nurses on site, the proposal by Grange [23] was most accurate, with only 7.14% under the actual number of nurses on site. The proposal by the PRIMA model was 14.3% higher than the exact number of nurses on site. The proposed number of nurses by Lund [11] was 3.14 times higher than the actual number of nurses on site, and the proposal of Arbon was 78.6% lower than the exact number [22].

When proposing the optimal number of first aid responders (or EMS providers), Lund [11] was closest to the actual number by suggesting 15.8% more than the exact number of first aid responders on site. The proposal of PRIMA was 40.4% lower than the actual number of onsite first responders. The proposed number of first aid responders by Grange [23] was 88.6% lower than the exact number of first aid responders on site.

### 3.5. Impact on Regular EMS

In 2017 and 2018, the ambulance of Boom had a mean number of EMS calls of 2.43 and 1.88 per 24 h. During weekend days (Friday, Saturday, and Sunday) in June, July, and August. On average, the ambulance of Boom did not have more EMS calls during the period of the MG (*M* = 3.83; *SE* = 0.47) than during the weekend days when there is no MG (*M* = 2.39; *SE* = 0.210). The difference, 1.44, (95% CI [−0.23 to 3.11]), was not significant, *t* (3.07) = 2.53, *p* = 0.068. Although there is a slight increase in the number of EMS calls, at no point in time during the event did the ambulance of Boom exceed its standard operational capacity.

In 2017 and 2018, the ambulance of Willebroek had a mean number of EMS calls of 5.24 and 5.46 per 24 h. During weekend days (Friday, Saturday, and Sunday) in June, July, and August. On average, the ambulance of Willebroek had fewer EMS calls during the period of the MG (*M* = 4.67; *SE* = 1.06) than during the weekend days when there is no MG (*M* = 5.34; *SE* = 0.39). This difference, −0.67 (95% CI [−3.94 to 2.66]), was not significant *t* (3.13) = 0.599, *p* = 0.589.

### 3.6. Impact on ED Visits and Hospital Admissions

During the event, the five hospitals closest to the event had 207 ED admissions from the festival or camping site, with 152 (73.43% [95% CI 4.04 to 56.76]) being brought in by IEHS. Thirty-two patients (15.46% [95% CI −3.30 to 16.10]) were admitted to hospital, with 19 patients (9.18% [95% CI −6.73 to 14.33]) admitted to the intensive care unit (ICU). On average, the EDs did not have more ED visits during the MGE (*M* = 82.03, *SE* = 3.246) than on days without the MGE (*M* = 88.38, *SE* = 4.20). This difference, −6.343 (95% CI −16.98 to 4.29), was statistically not significant, *t*(1.195) = 54.54, *p* = 0.2373, and there was no effect (r = 0.03). None of the EDs ever exceeded its standard operational capacity.

Regarding hospital admissions, on average there were no more hospital admissions during the MGE (*M* = 19.43, *SE* = 1.275), than on non-MGE days (*M* = 22.43, *SE* = 1.112). This difference, −2.997 (95% CI −6.384 to 0.391), was statistically not significant, *t*(1.771) = 56.96, *p* = 0.0819, and there was no effect (r = 0.05). As for admissions to the ICU, on average there were more additional admissions to ICU during the MGE (*M* = 2.067, *SE* = 0.332) than on days without the MGE (*M* = 1.485, *SE* = 0.225). This difference, 0.5817 (95% CI −0.2248 to 1.388), was statistically not significant, *t*(1.449) = 48.83, *p* = 0.1536, and there was no effect (r = 0.04).

In total, 18 (11.8% [95% CI 11.7 to 11.9]) patients who were taken to the hospital by IEHS remained admitted in the hospitals mentioned above for >24 h.

## 4. Discussion

We aimed to describe if the amount and level of onsite in-event health providers based on the PRIMA model mitigated the impact of Europe’s largest EDMF on local EMS and emergency departments. To our knowledge, we are the first to link patient data collected at an event to EMS and admissions in EDs to try to understand how EMS and EDs are impacted by MGs. The results of this study highlight several patterns that merit further attention.

With a PPR of 31.1/1.000 attendees, this rate is much higher than previous studies at EDMF, where PPR ranged between 1.45/1.000 attendees [6] and 20.8/1.000 attendees [12]. A possible explanation, we believe, is twofold. First, the majority of all patients seen presented with minor injuries (blisters, abrasions) or headaches. These injuries and headaches can be explained by the enthusiastic crowd, often dancing on unsuitable footwear for the environment and the omnipresent loud music and alcohol use leading to dehydration. Second, for most of the visitors, the presence of IEHS is another service provided by the event organizer, making the threshold for using these services low.

Most patients presented themselves to IEHS between 16:00 h and 18:00 h, coinciding with daily peak temperatures. This is earlier than previous studies reported [24]. When looking at the time high-acuity patients are brought in for medical help, two peak moments are witnessed. One between 21:00 h and 22:00 h, coinciding with one hour after sunset, which is in line with existing literature [23]. The second peak between 00:00 h and 01:00 h, coinciding with the closing hour of the festival and probably explained by the fatigue and alcohol or illicit drug use taking its toll, is not previously reported in mass gathering literature.

With an event-to-host-population ratio of 12.12, it is obvious the regular health infrastructure needed to be augmented. Similar to Locoh-Donou [25], we found that high severity presentations were rare. Despite this finding, we must argue that the number of patients treated and released by IEHS back to the event would have seriously disrupted the standard operational capacity of local EDs. With blisters and headaches being the most frequent complaint, a critical role lies with in-event first aid care workers (e.g., first-aid responders and paramedics). However, we can assume not all patients with blisters or headaches would have visited local EDs.

When looking at the higher acuity patients, we shift our focus to in-event professional healthcare workers (emergency nurses, emergency physicians). One out of three patients who needed in-event professional care (or even ALS) was intoxicated with illicit substances and/or alcohol, confirming existing literature [3,8,11,26,27]. Contrary to patients with minor acute medical needs, patients who needed in-event professional care presented to IEHS during the evening and night. Not needing all professional care staff throughout the event can be an essential consideration when creating work schedules.

With only 12.5% of all patients seen by in-event health professionals being transported to a hospital, the vast majority could return to the event—albeit after some medical supervision in the medical posts. The same pattern repeats itself when looking at those patients with an initially life-threatening condition (e.g., unconscious, compromised airway, etc.). Only 35 out of the initial 120 patients needing immediate medical care were required to be transported to a hospital. We assume the presence of in-event health professionals (along with prerequisite supplies) facilitated a hold-and-treat model of care, mitigating the effects of a more than a 12-fold increase in population on local health services. This result is similar to other studies [12,23].

Another essential consideration worth highlighting is that, despite the high PPR, onsite IEHS prevented excessive transport resources, easing the burden on regular EMS. Only 152 patients (1.22% of all patients seen and treated) needed transport to local hospitals, with only 35 patients (0.28% of all patients seen and treated) needing accompanying by an emergency physician and emergency nurse. This TTHR of 0.38/1.000 attendees is lower than previous publications, with Fitzgibbon reporting a TTHR of up to 2.1/1.000 attendees [3], but similar to the TTHR of 0.35/1.000 attendees reported by Turris [13]. If regular EMS services had had to transport all 152 patients to local hospitals, it would have caused significant disruption of normal EMS and health services.

With most Belgian EMS personnel trained at the BLS level, the number of transports of patients needing ALS support during transport (because of their altered or depressed levels of consciousness or compromised airways) would have had severe implications for local EMS. Depleting regular EMS would have caused severe disruption of EMS in the event region, with other ambulances having to respond to calls that otherwise would have been dispatched to the ambulances of Boom or Willebroek. In addition, depletion of medical and nursing staff from ED’s for transport accompaniment of critical patients from the MG would lead to increased crowding and congestion and a longer length of stay.

Finally, the statistically non-significant rise in EMS calls for the ambulance of Boom may be explained because they strengthen their availability to a 24/7 service during the event. The strengthening to a 24/7 service of the ambulance of Boom could also explain the statistically non-significant decline in EMS calls for the ambulance of Willebroek. However, the net effect is that the event’s impact on regular EMS is minimal.

The overall impact of the MGE on nearby hospitals was limited. Only the smaller general hospital Rivierenland in Rumst (only three kilometers from the event and camping site) received 114 patients from the MGE on their ED over the whole of the event period. With most patients only needing X-rays and nobody requiring admission to an intensive care unit (ICU), 105 patients could re-attend the MGE. Although the University Hospital Antwerp received 72 entries to their ED during the MGE period, with 19 patients admitted to ICU, causing a statistically significant rise in ICU admissions, only five patients remained hospitalized for longer than 24 h. Their ICU department never exceeded its standard operational capacity. With 17 admissions for the general hospital in Mechelen and two times two admissions for the Antwerp hospitals (GZA Sint-Augustinus hospital and ZNA Middelheim hospital), respectively, over the entire event period, the standard operational capacity of the emergency services were not jeopardized anywhere.

Several limitations should also be pointed out in the current study. Our study, like others, was limited by the lack of data of sufficient quality. Not using a data dictionary or minimal data set by the IEHS—as proposed by Ranse and Hutton [28]—leads to inconsistencies and variation in the level of data gathering. Another shortcoming of our study is that we do not know whether staffing (especially emergency nurses and emergency physicians) and accompanying costs with this staffing level, based on the PRIMA model advice, were optimal. Could more or less onsite professional staff have provided the same or even a better response? However, we assume that fewer onsite physicians would have resulted in more hospital transports, possibly overwhelming local EDs. A third limitation is that we did not collect data on patients referred to EDs but went by their means. These patients could have visited local EDs near the event site, but could well have reported to EDs near their home address. Therefore, we need to assume we underestimate the real impact of the MGE on local EDs. A fourth and final shortcoming was that we conducted the study at just one EDMF. However, with most of our results consistent with previous publications on EDMF, our results could be generalizable to similar EDMF but less to other MGE.

We recommend creating a catalog and timeline of advanced interventions performed on site and the rate of advanced interventions per qualified provider per hour to see if staffing levels are an optimal and effective means of mitigating ambulance and ED usage in further research. A cost analysis can substantiate this to see whether the optimal number and staffing level are cost-effective.

## 5. Conclusions

This study suggests the reduced need for patient transports because of the onsite availability of skilled personnel with adequate medical material, stemming from the predictions of the PRIMA model. This study demonstrated that although there is an impact on EMS, IEHS, and EDs of local hospitals from MGEs, the adopted hold-and-treat model of care with in-event health professionals mitigated a surge in interventions of the host community EMS and helped to reduce the overall burden on local EDs by providing adequate onsite care. Further research on the impact of MGEs on regular emergency health services (e.g., EMS, EDs) and optimal staffing levels are needed to provide a greater understanding of the health outcomes for MGE attendees and the host community.

## Figures and Tables

**Figure 1 ijerph-20-03207-f001:**
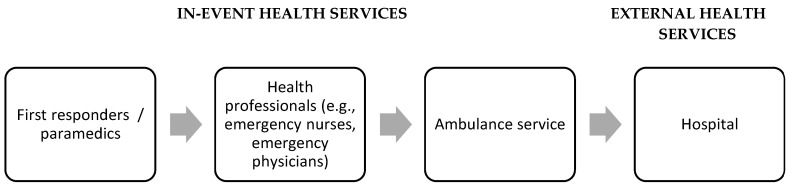
Hierarchy of Clinical Care at Belgian Mass Gatherings (Modified from Ranse et al., 2017 [4]).

**Figure 2 ijerph-20-03207-f002:**
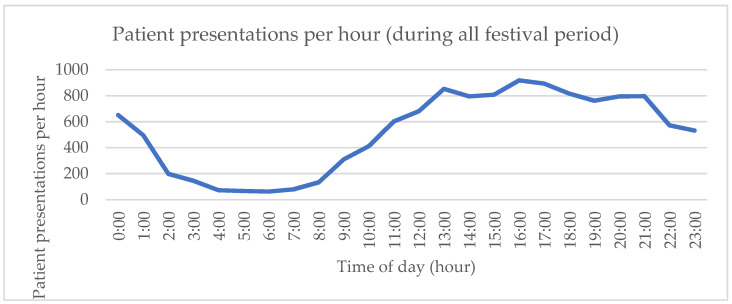
Patient presentations per hour to IEHS.

**Figure 3 ijerph-20-03207-f003:**
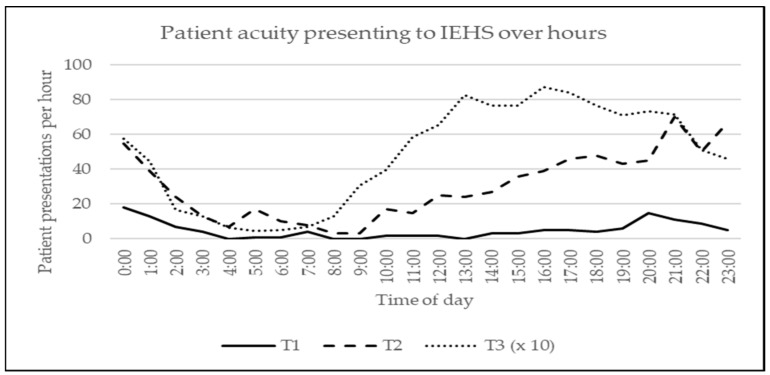
Patient acuity presenting to IEHS over hours (the number of T3 patients was divided by 10).

**Table 1 ijerph-20-03207-t001:** Advice on in-event health services staffing by the PRIMA model.

Camping Site (*)	Festival Site (†)
3 medical posts	3 medical posts
1 mobile urgency group (1 emergency physician; 1 emergency nurse)	1 mobile urgency group (1 emergency physician; 1 emergency nurse)
3 emergency physicians	6 emergency physicians
5 emergency nurses	11 emergency nurses
3 ambulances (2 paramedics/ambulance)	6 ambulances (2 paramedics/ambulance)
1 BLS crew (3 first aid responders)	14 BLS crews (3 first responders/crew)
21 first aid responders	47 first aid responders

* Opening hours medical posts on camping site: Thursday: 12:00 h–Monday: 16:00 h. † Opening hours medical posts on the festival site: Friday: 11:00 h–01:00 h. Saturday: 11:00 h–01:00 h. Sunday: 11:00 h–00:00 h.

**Table 2 ijerph-20-03207-t002:** Meteorological data on both MG weekends.

Date	Temperature (°C)			Average Humidity (%)	Sunset
**Weekend 1**	**12:00 h**	**18:00 h**	**24:00 h**		
Thursday, 18 July 2019	21	18	14	72%	20:49 h
Friday, 19 July 2019	23	24	16	68%	20:48 h
Saturday, 20 July 2019	23	24	19	79%	20:47 h
Sunday, 21 July 21 2019	23	22	17	70%	20:46 h
Monday, 22 July 2019	25	27	18	55%	20:45 h
**Weekend 2**					
Thursday, 25 July 2019	36	38	24	50%	20:40 h
Friday, 26 July 2019	27	25	22	73%	20:39 h
Saturday, 27 July 2019	24	21	20	81%	20:38 h
Sunday, 28 July 2019	18	17	13	91%	20:37 h
Monday, 29 July 2019	24	26	17	64%	20:36 h

**Table 3 ijerph-20-03207-t003:** Comparison between weekend 1 and weekend 2.

Weekend 1			Weekend 2			Total		
Presentations	*n*	(%)	Presentations	*n*	(%)		N	(%)
6165			6286			12,451		
**Gender**			**Gender**			**Gender**		
Male	3326	(53.9)	Male	3278	(52.1)	Male	6604	(53.04)
Female	2839	(46.1)	Female	3008	(47.9)	Female	5847	(46.96)
**Triage Code ^a^**			**Triage Code ^a^**			**Triage Code ^a^**		
T1	58	(0.9)	T1	62	(1.0)	T1	120	(0.96)
T2	286	(4.6)	T2	445	(7.1)	T2	731	(5.87)
T3	5821	(94.4)	T3	5779	(91.9)	T3	11,600	(93.17)
**Level of Care ^b^**			**Level of Care ^b^**			**Level of Care ^b^**		
In-event health professional care	344	(5.6)	In-event health professional care	507	(8.1)	In-event health professional care	851	(6.83)
In-event first aid care	5821	(94.4)	In-event first aid care	5779	(91.9)	In-event health first aid care	11,600	(93.17)
**Discharge Disposition**			**Discharge Disposition**			**Discharge Disposition**		
Return to MG	6082	(98.7)	Return to MG	6200	(98.7)	Return to MG	12,282	(98.64)
Transport to hospital by own means	8	(0.1)	Transport to hospital by own means	9	(0.1)	Transport to hospital by own means	17	(0.14)
Transport to hospital with IEHS	75	(1.2)	Transport to hospital with IEHS	77	(1.2)	Transport to hospital with IEHS	152	(1.22)

^a^ Triage codes: T1: immediate (these patients need immediate resuscitative interventions for survival—seen by in-event health professional care; e.g., airway obstruction); T2: urgent (require early treatment—seen by in-event health professional care; e.g., limb fractures); T3: minor (ambulatory patients who follow commands—seen by in-event first aid care; e.g., soft tissue injury). ^b^ Level of care: In-event health professional care: all T1 and T2 patients seen by health professionals (emergency nurses and emergency physicians); In-event health first aid: all T3 patients seen by first aid responders and paramedics.

**Table 4 ijerph-20-03207-t004:** Pathology seen by in-event health professionals.

	Frequency	Percent	Cumulative Percent	95% Confidence Interval	
				Lower	Upper
Intoxication (alcohol/drugs)	280	32.9	32.9	29.6	36.0
Heat/cold related problem	54	6.4	39.3	4.3	8.6
Syncope	44	5.2	44.5	3.8	6.8
Gastro-intestinal complaint	42	4.9	49.4	3.5	6.7
Other illness	42	4.9	54.3	3.6	6.4
Other trauma	38	4.5	58.8	3.2	5.8
Musculoskeletal problems	36	4.2	63.0	1.2	7.3
Headache	25	2.9	65.9	1.9	4.1
Chest pain	24	2.8	68.7	1.7	3.9
Wounds	23	2.7	71.4	1.7	3.9
Respiratory distress	21	2.5	73.9	1.5	3.6
Hyperventilation	15	1.8	75.7	0.9	2.7
Allergy	9	1.1	76.7	0.5	1.8
Sore throat	7	0.8	77.5	0.2	1.5
Eye injury	6	0.7	78.2	0.2	1.3
Insect bite	5	0.6	78.8	0.1	1.4
Blisters	4	0.5	79.3	0.1	1.0
Aftercare illness	4	0.5	79.8	0.1	1.0
Aftercare injury	2	0.2	80.0	0.0	0.6
Burns	1	0.1	80.1	0.0	0.4
Missing patient details	169	19.9	100.0	17.2	22.7
Total	851	100.0	100.0		

## Data Availability

The data presented in this study are available on request from the corresponding author. The data are not publicly available due to privacy reasons.

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
