# Peer review of "The Impact of In-Event Health Services at Europe’s Largest Electronic Dance Music Festival on Ems and Ed in the Host Community"

_ijerph, 2023, doi:10.3390/ijerph20043207_

Round 1

Reviewer 1 Report

Thank you so much for the opportunity to review this excellent manuscript. It was an engrossing read! I will be excited to see it published and look forward to citing your work in the near future.

My questions/comments are below. None of the changes I suggest are substantive. 

PRIMA

  1. Is this model available/accessible for use by other researchers? 

  2. Is the model on a website that could be cited in this manuscript? I know readers will want to know more!

  3. On Page 2, Line 56, what does “sufficiently valid” mean? Are you saying the model is rigorous and reliable?

Demographics of the Event and the Patients

  1. Number of attendees (gate) versus number of tickets sold. I understand that there were 400,000 tickets sold. Was that 400,000 tickets sold (every ticket for the entire festival)? This would mean a potential 400,000 unique attendees. Or, were the tickets for specific days (and so a smaller number of unique individuals)? 

  2. What was the actual number of days for the event? You nicely describe the Friday-Monday event schedule (8 days). However, in the results section of the manuscript you use 10 days as the number of days for the event. I imagine this includes the load in day for those who are camping?

  3. Please choose whether you wish to use the term “gender” or the term “sex.” You have used both in this manuscript. I believe you mean sex (and there were only two options (male, female). In the narrative you use “gender” and in the table you use “sex.” (Side question… Will The Flemish Cross eventually include more selections for gender?)

Grammar, Punctuation & Assorted

  1. In full disclosure, I speak no language fluently with the exception of English (and there are some who would argue about that). For this manuscript, I would recommend the following:

    1. Commit to past tense throughout the document.

    2. Do a careful read for grammar, spacing and punctuation. Examples: 

      1. Please break the first sentence of the manuscript into two sentences.

      2. Decide whether you are using “MGE” to mean a “mass gathering event” OR “mass gathering events.” (Used both ways on Page 2).

      3. Consider including a table of abbreviations if permitted by the journal.

      4. Page 2, Line 50, consider using the phrase “skill set” versus “ability.”

      5. On Page 2, Line 59, there is a comma that can be removed.

      6. On Page 2, Line 80, this might read more clearly as “one gate was for…”

      7. Would you consider using the 24-hour clock for the times? For example, on Line 68, 0100 hours would be very clear (or 1 am).

      8. On Page 4, Line 165, the text should read “as an additional file.”

      9. On Page 4, Line 136, I believe there is a “t” missing on the word “even.”

      10. On Page 6, Line 263, does “EC” stand for ethics committee? If so, would you please avoid using the abbreviation?

      11. Page 8, Line 310, should read “went to the hospital by their own means.”

      12. Line 330 should read “hospital by their own means.”

      13. On Page 15, Line 564, there is a question mark that should likely be a “.”.

    3. Would you consider adding “Event Medicine” to the keywords? This might help with future retrieval of your wonderful work.

    4. As the ethics section appears in both the body of the manuscript AND as a footnote at the end of the manuscript, you could substantially trim the narrative in the body of the manuscript (and refer your readers to the end of the paper for additional details).

On Page 3, Line 99, you make a comment about 1 ambulance for every 21,000 attendees. Is this regardless of event type?

On Page 3, Line 120, you comment about nurse and physician utilization rates. Would you be able to clarify the source of the numbers? Are the numbers drawn from previous studies you have conducted?

Your description of the Belgian health-care system (and the resources available regionally) was so well done. I was interested to read that the in-event health care services were responsible for transporting patients from the event to the hospital. Where I live, no patients can be transported except by government-supported vehicles.

The Tables are lovely. It is very easy for the reader to digest the content.

Are you able to add brief content about how you obtained information from the surrounding hospitals? Such information would be very difficult to obtain in my practice setting. Did you have monitors in the ED? Did you have access to patient records? Was there a question added to the triage forms in the ED such that the festival was named as part of the demographics of the ED visit?

You might consider briefly introducing the “hold and treat model” at the beginning of the manuscript so that when you mention the model at the end of the paper, the dots are connected for your reader.

Author Response

Thank you for your kind words and your beneficial review. We are most grateful for your time providing us with suggestions on improving our manuscript. In our revision, we have addressed your recommendations as well as possible and specified them in detail in the document as an attachment.

Reviewer 2 Report

Thank you for the opportunity to review your paper. Overall, this paper is well-written and makes a valuable contribution to the literature in this space. Advancing our understanding of the impacts on health services from mass gathering events is important, reporting variables such as: event-to-host population ratio is a good example of where this paper builds on existing scientific understandings of this impact.

Consider changing all 'MG' to 'MGEs' throughout the paper.

1. What is the main question addressed by the research?

What is the impact on emergency health services from an electronic dance music festival.

2. Do you consider the topic original or relevant in the field? Does it
address a specific gap in the field?

Yes, there is only 1 or 2 other papers in the world that address this issue.

3. What does it add to the subject area compared with other published
material?

New knowledge, and additional perspectives such as the event-to-host population ratio.

4. What specific improvements should the authors consider regarding the
methodology? What further controls should be considered? 

Nil

5. Are the conclusions consistent with the evidence and arguments presented and do they address the main question posed?

Yes. These are appropriate.

6. Are the references appropriate?

Yes

7. Please include any additional comments on the tables and figures.

Nil

Author Response

Thank you for your kind words and your beneficial review. We are most grateful for your time providing us with suggestions on improving our manuscript. In our revision, we have addressed your recommendation as well as possible and specified it in detail in the document as an attachment.

Reviewer 3 Report

I Thank the author's for his article. This subject is an interessant topic about Emergency Overcrowding in the context of an mass event and impact of local emergency unity to emergency transfer. Obviously i have some remark about methodology and results. on this paper I have serious reservations about its publication. The model presented (PRISMA) is not evaluated in its performance but only presented in a descriptive way. The design of the study, which could be a before and after, is essential to demonstrate the usefulness of the intervention presented. The authors could otherwise present the performance of the prediction in relation to reality, which is not the case here. As it stands, I can only recommend rejection of the publication with a proposal to resubmit it subject to major improvements to the paper. 

In first, and general comment, i’m desappointed about lack of nationwide data about emergency sollicitation who should be used like control group. It would be intressant to use longitudinal data to contextualise the theorical trend of emergency use situation.

Major Comment

Introduction is clearly writting but should be improve by a better contextualisation of hypothesis. This study should be a before after study who can evalue impact of the implementation of an local emergency unit. 

Method

It is a retrospective study but authors don’t precise if it’s crossway design or before after. A cross way design is a several lack and i recommand to compare with a before after design (with several limits). A better approach could associate a propensity score calculate on emergency consumption, user caracteristics and other during the festival period in a retrospective design. 

Study design describe context of festival it is innapropriate in this context. It could be describe in population paragraph. 

It is not clear if this study compare camping site to festival site or other. Table 1 is not clear about this assumption. This study could be improve if they compare in (at minima) before after design the situation before and after application of PRISMA model in festival. 

Impact on hours on outcome is not clearly mentionned. It is not clear about hours. Does night hours impact quality of care or capacity of emergency system to transfer patient in an appropriate hospital ? 

In statistical method, there is no precision about adjustement on Odds ratio. It could be aids the readers about the performance of the model. 

In Results 

A flowchart could be help the reader to a comprehensive approach about data who were analyzed

In table 5. Results about comparison between model could be evaluate (i.e emergency transfer /hours) it is not clear. 

It is non necessary to purpose in result comparison with other model, it could be placed on discussion

Presentation by hospital is not efficient and clear. 

Figure with temperature could be agregate in one (fig 2 and 3)

Discussion

First part of discussion do not conclude to hypothesis, it could be more clear than PRISMA model or intervention demonstrate a clear effect on emergency transfer or access. 

Minor Comment 

Line 74 MG or MGE ? 

Author Response

Thank you for your review. We are most grateful for your time providing us with suggestions on improving our manuscript. In our revision, we have addressed your recommendations as well as possible and specified them in detail in the document as an attachment.

Round 2

Reviewer 3 Report

I would like to thank the authors for answering some of my comments. 

However, I think the authors have misunderstood my concerns about the article. Like the other Reviewers, I think that the article presents original and interesting data. However, independently of this, I indicated that the methods do not accurately describe the type of study that was conducted. This is a major problem, and needs to be corrected. 

The authors write (l. 65 p. 2): "This retrospective descriptive study was conducted at one 65 outdoor EDMF during two weekends in July 2019” 

This study evaluates the association of the presence of a festival (we can call this a risk factor, as in increasing a risk of disease – due to alcohol, crowding etc) with the occurrence of Emergency Department presentations (which would be a studied outcome). 

Analytic studies estimate the relationship between a risk factor and an outcome. Here, the risk factor was not present in the first period (no festival), then it is present in the second period (festival). Therefore, this study is, quite evidently, a before-after study. It should be described as such. Also, it is ok if there are more emergency presentations during the festival. This is to be expected. The key point to consider (which is also the aim of the study) is the strenght of the association (effect size). 

Here are other points to consider (some of them relate to the above), presented following the article order: 

Abstract: 

Methods reported here are not specific enough, in addition to saying what tests were made there needs to be at least an outline of the main variable studied (what separated the groups during comparisons). 

Main article text: 

Introduction: The citation by Fitzgibbon is not a “definition”, rather a description. 

Please add the meaning of “egress” as this word is not often used in everyday language. 

« As a recent study showed, the Belgian Plan Risk Manifestation (PRIMA) model is a 55 valid prediction tool for predicting patient presentations at music MGEs in Belgium [9]. » : it seems to be more than this, the prediction model seems to be accompanied by a staffing recommendation plan. 

l.60 p.2: “has hardly been studied » : sentence needs to be reformulated 

The aim of the study or the methods need to be reformulated. The methods later say that the study is descriptive, if so the verb in the aim should be « to describe … ».  

during two weekends in July 2019 in Boom, Belgium, with approximately 18,000 inhabitants” : sentence needs to be reformulated 

The authors should use past tense for events that occurred in the past throughout the article: 

« tickets are sold out » -> « tickets were sold out » (but the authors consider removing the sentence instead as it does not contribute to answering the study question), “potable water WAS freely available” …  

« Belgian Federal Public Health Service based » : there needs to be a comma between Service and based. 

« All medical posts at the festival site had at least thirty-eight first-aid responders, paramedics » if this figure is correct 38 should be written in number form. Also, the authors should add the precision that it was 38 per post or 38 after summing all posts of the festival. If there were only two physicians and paramedics, the authors could say what the other staff were (role and/or general qualifications to provide first aid, how were they trained). 

« . On average, the ambulance in Willebroek has a mean number of 5.30 187 EMS responses on weekend days » at this stage we don’t know what Willebroek is and why it is mentionned 

« Data for this study was handed to the lead investigator in a MS Excel 2019® spreadsheet » : repetition 

« We did not encounter missing data, as only IEHS had missing data » could be rephrased as there seems to be an apparent contradiction. 

« (the most resistant statistic) » that clause can be removed 

« We calculated the odds ratios to determine whether being treated by first aid personnel or health care professionals was a predictive variable for the need to transport to 259 the hospital. » this sentence needs to be rephrased to emphasise the comparison being made (using the term « versus » for example) 

Results : 

Figure 3: The time period referred to here should be written also in figure title or legend : « Sum of all same hour presentations during all festival period (10 days) » for example.  

Table 4:  « valid percent » could be removed (repetition) 

Fig 3 : the authors could consider using log scale to avoid mismatch in y scale across groups. 

l.363 p.10 : This comparison could be removed altogether. The MGE organisators will have used one or more of these models to determine staffing. Therefore, the difference between predictions and reality do not give much useful information, and staffing is strongly expected to be similar to predictions +- random statistical noise due to human ressources issues. 

The authors should focus more on the outcome side 

l.398 p.11 : There probably should be an overall analysis of all hospital admissions for all sites (MGS vs non-MGE). Then the analysis for individual hospitals should go into a Supplementary Material or appendix. The breakdown by hospital artificially reduces statistical power. The authors could also comment on the effect size of the differences which could be low for the overall analysis, despite a statistically significant difference. 

Discussion : 

« following Locoh-Donou » : a number extracted from cited litterature that supports the claim that that this follow that article should be provided. 
